# The Role of Aldosterone and the Mineralocorticoid Receptor in Metabolic Dysfunction-Associated Steatotic Liver Disease

**DOI:** 10.3390/biomedicines13081792

**Published:** 2025-07-22

**Authors:** Mohammed Barigou, Imran Ramzan, Dionysios V. Chartoumpekis

**Affiliations:** 1Department of Diabetes, College of Health Sciences, University of Doha for Science and Technology, Doha 24449, Qatar; imran.ramzan@udst.edu.qa; 2Division of Endocrinology, School of Medicine, University of Patras, 26500 Patras, Greece; dchart@upatras.gr

**Keywords:** MASLD, MASH, fibrosis, aldosterone, mineralocorticoid receptor, SGK1, TGFβ1

## Abstract

Metabolic Dysfunction-Associated Steatotic Liver Disease (MASLD) is one of the fastest-growing hepatic disorders worldwide. During its natural course, MASLD tends to progress from isolated steatosis of the liver to Metabolic Dysfunction-Associated Alcoholic Steatohepatitis (MASH), advanced fibrosis, and finally cirrhosis, with the risk of developing hepatocellular carcinoma (HCC). Although frequently related to overweight or obesity and other components of the metabolic syndrome (MS), MASLD can also be present in individuals without such risk factors. The mechanisms leading to MASLD are incompletely elucidated and may involve many proinflammatory and pro-fibrotic pathways, disrupted biliary acid homeostasis, and gut microbiota dysbiosis. Aldosterone and its interaction with the mineralocorticoid receptor (MR) are thought to participate in the pathogenesis of MASLD through the modulation of inflammation and fibrosis. Remarkably, blockade of the MR in experimental models was shown to improve MASH and fibrosis through mechanisms that need further characterization. So far, however, few clinical studies have explored the effect of MR blockade in the management of MASH and associated fibrosis. This review is intended to summarize the recent animal and human data concerning the interaction between MR pathways and MASLD.

## 1. Introduction

Metabolic Dysfunction-Associated Steatotic Liver Disease (MASLD), characterized by excessive fat accumulation in hepatocytes (greater than 5% of liver weight), has emerged as a major global cause of chronic liver disease [1]. Its prevalence is predicted to rise along with increasing rates of obesity, type 2 diabetes, and metabolic syndrome (Figure 1) [1]. MASLD has been associated with increased liver- and cardiovascular-related mortality [2]. Predictive statistical models estimate that MASLD will become the leading cause of liver transplantation worldwide by 2030 [3,4]. Recent guidelines issued by the European Association for the Study of the Liver (EASL) recommend a systematic screening for MASLD in individuals with a high metabolic risk using noninvasive biomarkers such as the MASLD fibrosis score (NFS; a composite score including age, hyperglycemia, body mass index, platelet count, albumin, and aspartate and alanine aminotransferase concentrations), potentially complemented with transient elastography (Fibroscan^®^) to identify individuals suspected to have advanced fibrosis [5]. Yet, liver biopsy remains the gold-standard method for the diagnosis of MASH and related hepatic fibrosis [5]. However, despite the availability of several guidelines and consensus statements, as well as a large number of clinical trials, there are no established drug therapies for MASLD, and its clinical management remains a challenge [5].

The strong link between insulin resistance and MASLD has been well established in the literature and is considered a central feature in the pathogenesis of this condition. However, emerging evidence suggests that additional signaling pathways beyond insulin resistance may significantly contribute to the onset, progression, and metabolic complications of MASLD. Among these, the renin–angiotensin–aldosterone system (RAAS) has garnered increasing attention as a potential key player in hepatic metabolic dysfunction and fibrogenesis [6,7]. Notably, some clinical conditions characterized by persistent overactivation of the RAAS—including primary aldosteronism (PA), Cushing’s syndrome, and polycystic ovary syndrome (PCOS)—are commonly associated with features of metabolic syndrome and show a disproportionately high prevalence of MASLD in affected individuals [8,9,10]. Despite these associations, the underlying mechanisms linking RAAS overactivity to MASLD development in these endocrine disorders remain incompletely understood. It is likely that MASLD in these contexts arises from a complex and multifactorial interplay between insulin resistance and hormonal imbalances involving pathways such as RAAS signaling and activation of the mineralocorticoid receptor (MR) [10,11,12,13,14]. In particular, increased RAAS activity has been shown to exacerbate systemic insulin resistance and promote endothelial dysfunction and oxidative stress, both of which may further aggravate hepatic steatosis and inflammation [14]. These effects suggest that RAAS hyperactivity may contribute to MASLD pathogenesis not merely as a secondary metabolic consequence but as a potential primary driver through non-insulin-mediated mechanisms. As such, dissecting the contribution of RAAS and MR signaling in MASLD could offer novel insights into disease heterogeneity and may help identify subgroups of patients who could benefit from targeted hormonal therapies alongside conventional lifestyle and metabolic interventions [14].

Figure 1 emphasizes the natural history of MASLD and highlights the protective and risk factors implicated in the disease’s evolution.

## 2. Mechanisms Involving Aldosterone and MR in MASLD

### 2.1. General Mechanisms of Aldosterone and Mineralocorticoid Receptor (MR)- Induced Inflammation and Fibrosis

Aldosterone is a steroid hormone synthesized in the zona glomerulosa of the adrenal cortex through a multi-step enzymatic process involving cholesterol side-chain cleavage and sequential conversions mediated by enzymes such as 3β-hydroxysteroid dehydrogenase and aldosterone synthase (CYP11B2) [8,15]. Its secretion is primarily regulated by the renin–angiotensin–aldosterone system (RAAS), serum potassium levels, and adrenocorticotropic hormone (ACTH) [13]. While aldosterone is classically viewed as an endocrine hormone acting on renal epithelial tissues, recent evidence suggests that under pathological conditions, such as chronic inflammation and fibrosis, aldosterone may also be synthesized locally in peripheral tissues, including the liver [7,8,11,12]. This extrarenal or paracrine aldosterone production may enhance localized mineralocorticoid receptor (MR) activation, particularly in the context of MASLD and hepatic fibrogenesis [11,12,13]. 

The mineralocorticoid receptor (MR) itself is a nuclear hormone receptor that exhibits a high affinity for both mineralocorticoids and glucocorticoids, and it is expressed in a variety of tissues throughout the body. MR activation is recognized to exert potent proinflammatory and pro-fibrotic effects, which have been well-documented across numerous experimental models and in several different tissue types, including the heart, kidneys, vasculature, and liver [15,16,17]. The molecular mechanisms through which MR mediates inflammation and fibrosis involve several key signaling molecules, notably serum- and glucocorticoid-inducible kinase 1 (SGK1) and transforming growth factor β (TGF-β). Both of these downstream effectors are transcriptionally upregulated in response to mineralocorticoid binding and are activated through a signaling cascade dependent on phosphatidylinositide 3-kinase (PI3K) activity [18,19,20]. SGK1 plays a particularly central role in amplifying inflammatory responses by enhancing the activity of the nuclear factor-κB (NFκB) pathway, a critical transcription factor known to regulate the expression of a broad array of inflammatory cytokines and mediators, including connective tissue growth factor, a key player in fibrosis [18,21,22]. In addition to its role in promoting inflammation, SGK1 also contributes to fibrosis progression by stabilizing and preventing the degradation of Smad2/3, transcription factors activated by TGF-β signaling that are essential for collagen deposition and extracellular matrix remodeling [23]. Elevated expression levels of SGK1 have been observed in a wide spectrum of chronic fibrosis disorders, including MASLD, further supporting its pathological relevance in hepatic fibrosis and metabolic inflammation [24,25,26]. Encouragingly, the deleterious effects associated with mineralocorticoid excess—namely, inflammation and fibrosis—can be significantly attenuated, and in some cases even reversed, through pharmacological inhibition of MR. As such, MR antagonists have emerged as promising therapeutic agents in the management of diseases characterized by chronic inflammation and tissue fibrosis, including MASLD and related cardiometabolic conditions [22,27,28]. These findings underscore the therapeutic potential of targeting MR signaling as a strategy to modulate key fibrogenic pathways and reduce inflammation-driven organ damage in a variety of clinical settings.

### 2.2. Mechanisms Involving Aldosterone and MR in Experimental Models of MASLD

It is now widely recognized that the physiological actions of mineralocorticoids extend far beyond their classical role in regulating blood pressure and electrolyte balance. Increasing evidence supports their involvement in a broader spectrum of biological functions, particularly in the promotion of inflammation and fibrotic tissue remodeling [24,25,26]. At the hepatic level, hepatic stellate cells (HSCs)—the principal effector cells in the development of liver fibrosis—have been shown to express MR [29]. In experimental models of MASLD, the administration of selective aldosterone antagonists has consistently demonstrated beneficial effects, including a marked reduction in HSC activation and proliferation. These effects are accompanied by the downregulation of TGF-β1 expression, suppression of proinflammatory cytokines such as TNFα and IL-6, decreased levels of plasminogen activator inhibitor-1 (PAI-1), reduced hepatic collagen accumulation, and a simultaneous increase in the expression of endothelial nitric oxide synthase (eNOS) [30,31]. These therapeutic benefits were also observed in murine models of MASH driven by the overexpression of sterol regulatory element-binding protein-1c (SREBP-1c), a central regulator of hepatic de novo lipogenesis. In these models, treatment with eplerenone—a selective MR antagonist devoid of antiandrogenic activity—led to significant histological improvements in liver architecture and inflammation [32]. Furthermore, genetic deletion of aldosterone synthase (CYP11B2) in mice conferred resistance to diet-induced hepatic steatosis, indicating that local aldosterone production may play a functional role in liver pathology under metabolic stress conditions [33]. Importantly, MR mRNA expression in HSCs was found to correlate strongly with fibrogenic and inflammatory gene expression profiles, reinforcing the pathological relevance of MR signaling in hepatic fibrosis [34]. Emerging data suggest that aldosterone synthesis may occur, at least partially, within the liver in experimental MASLD models. Upregulation of CYP11B2 mRNA has been detected in fibrotic rat livers exposed to carbon tetrachloride (CCl_4_), and MR blockade with spironolactone partially inhibited fibrogenesis during the early stages of CCl_4_-induced hepatic injury [35]. More recently, Yang Li et al. [36] demonstrated that aldosterone directly activates primary mouse HSCs by inducing the expression and assembly of the NOD-like receptor protein 3 (NLRP3) inflammasome. This multiprotein complex mediates the activation of inflammatory caspases and maturation of key cytokines like IL-1β and IL-18, thereby exacerbating inflammation and fibrosis. In this context, genetic knockout of NLRP3 ameliorated aldosterone-induced hepatic fibrosis, while spironolactone treatment inhibited NLRP3 activation and significantly reduced liver damage in fibrosis mouse models. In another study, Adeyanju et al. [37] investigated the hepatoprotective potential of spironolactone in a PCOS rat model induced by letrozole, an aromatase inhibitor. Treatment with spironolactone effectively reversed the hepatic accumulation of triglycerides and uric acid and significantly improved MASLD-related histological alterations, further supporting its role in reducing hepatic metabolic injury.

Finally, emerging data suggest that the mineralocorticoid receptor (MR) may influence the risk and progression of MASLD not only through direct hepatic mechanisms but also via its role in regulating immune responses along the spleen–liver axis. In fact, MR is expressed in T and B lymphocytes and has been shown to modulate splenic lymphocyte proliferation and inflammation [38]. In obesity-induced fatty liver, splenic enrichment of myeloid-derived suppressor cells (MDSCs) and natural killer T (NKT) cells correlates strongly with their hepatic counterparts, supporting the idea of a coordinated, cell-specific immune crosstalk between spleen and liver [39]. These findings may suggest an immunometabolic pathway through which MR signaling may indirectly exacerbate MASLD via modulation of the spleen–liver inflammatory axis [38,39].

Collectively, these experimental findings strongly support an independent and direct role of aldosterone and MR signaling in promoting hepatic steatosis, inflammation, and fibrosis. They provide compelling evidence that the MR blockade holds therapeutic promise for MASH and possibly broader MASLD phenotypes. However, while MR and aldosterone are clearly implicated in the pathogenesis and progression of MASLD in preclinical models, the specific intracellular pathways mediating these effects remain only partially understood. Among potential mediators, SGK1 overactivation has emerged as a key candidate. In rodent studies, the introduction of a gain-of-function SGK1 mutation intensified both hepatic steatosis and fibrotic responses, while CYP11B2 deficiency protected against MASLD in animals with constitutive SGK1 activation [25,40]. Furthermore, recent transcriptomic analyses have identified SGK1 as a crucial node in the PI3K-AKT signaling axis, linking it to both insulin resistance and MASLD pathogenesis [41]. These findings underscore the complex and multifaceted roles of MR-related pathways in metabolic liver disease and support the rationale for continued investigation into MR antagonists as a therapeutic strategy.

Figure 2 highlights the principal pathway by which MR activation induces inflammation and fibrosis in the liver:

### 2.3. Human Studies Linking Aldosterone and Mineralocorticoid Receptor to MASLD

#### 2.3.1. Observational Studies

The role of aldosterone and its interaction with MR has also been explored in human studies, particularly in clinical conditions characterized by systemic or organ-specific fibrosis. In humans, aldosterone has long been implicated in promoting both myocardial and renal fibrosis, processes that are typically ameliorated or even reversed with the administration of MR antagonists such as spironolactone or eplerenone [42,43,44]. These observations suggest that MR activation may have pathologic consequences beyond cardiovascular and renal tissues and could extend to hepatic fibrogenesis in the context of MASLD. In the Jackson Heart Study, a large-scale observational cohort comprising 2507 African American participants from Mississippi, investigators assessed liver fat content via computed tomography alongside serum aldosterone levels in individuals who were not receiving RAAS-inhibiting therapies. Both univariate and multivariate regression analyses identified a statistically significant association between elevated aldosterone concentrations and the presence of hepatic steatosis, particularly in African American women, highlighting a possible gender-specific vulnerability to aldosterone-related hepatic fat accumulation [45].

Moreover, individuals with PA have been consistently shown to carry a higher risk of metabolic disturbances compared to those with essential hypertension. Specifically, PA patients demonstrate a higher prevalence of metabolic syndrome and an increased occurrence of MASLD, even in the absence of traditional risk factors for chronic liver disease such as alcohol consumption or viral hepatitis [46,47]. Notably, among PA patients, those presenting with hypokalemia—an indicator of more pronounced aldosterone activity—are more insulin resistant and exhibit a disproportionately higher frequency of MASLD than normokalemic counterparts. This suggests that the severity of aldosterone excess may modulate the degree of hepatic and metabolic dysfunction [46].

In addition, various clinical conditions associated with heightened RAAS activation and increased aldosterone secretion, including PCOS and type 2 diabetes, have been linked to an elevated risk of MASLD [11,14,48]. These endocrine–metabolic disorders appear to share common mechanistic pathways that may involve MR signaling and chronic low-grade inflammation, contributing to hepatic fat accumulation and progression to steatohepatitis.

Further support for the role of aldosterone in liver pathology comes from a retrospective cohort study involving serum and liver tissue samples from 40 patients with and without histologically confirmed hepatic fibrosis. The study found that individuals with liver fibrosis exhibited significantly higher serum aldosterone levels, along with increased hepatic expression of NLRP3, a key inflammasome involved in fibrogenic signaling pathways [36]. These findings suggest a link between systemic aldosterone activity, hepatic inflammatory responses, and fibrotic remodeling.

Recently, a large-scale population-based cohort study examined the relationship between RAAS inhibition and the development or progression of MASLD in 212,146 individuals, including 27,565 patients with a prior diagnosis of MASLD. The study demonstrated that pharmacologic inhibition of the RAAS—through agents targeting various components of the pathway—was significantly associated with a reduced incidence and slowed progression of MASLD, particularly among individuals with a body mass index (BMI) ≥ 25 kg/m^2^ and normal fasting plasma glucose levels (<100 mg/dL) [12]. These results highlight the potential preventive value of RAAS-targeting strategies in metabolic liver disease, especially in overweight individuals without overt diabetes. These results were corroborated in a large, retrospective cohort study of hypertensive Chinese adults (n = 3713) in which higher plasma aldosterone concentrations (PAC) were independently associated with new-onset NAFLD. The authors found a non-linear (J-shaped) relationship, with significantly increased NAFLD risk when PAC reached ≥13 ng/dL; those in the highest tertile had ~1.7-fold greater odds versus the lowest tertile. This association persisted after adjusting for traditional risk factors. The results suggest that elevated aldosterone, common in hypertension, may directly contribute to hepatic steatosis via MR-mediated pathways and inflammatory/fibrotic mechanisms [49].

Finally, in a recent genetic association study involving over 125,000 participants from the UK Biobank, Zeng et al. [50] demonstrated that individuals with a high polygenic risk score (PRS) for renin-independent aldosteronism (RIA) had significantly increased odds of developing MASLD and cirrhosis. Mendelian randomization analysis confirmed a causal genetic link between RIA and these hepatic outcomes, with no evidence of pleiotropy or heterogeneity. These findings underscore the role of autonomous aldosterone excess as a genetically driven contributor to liver fat accumulation and fibrotic progression, independent of classic RAAS activation [50].

Taken together, these observational studies in human populations consistently support the notion that elevated aldosterone levels and enhanced MR signaling are linked to the development and severity of MASLD. While causality cannot be inferred from these studies, the accumulating evidence reinforces the rationale for further interventional trials to evaluate the therapeutic potential of MR blockade in MASLD.

#### 2.3.2. Interventional Studies

To date, only a limited number of interventional studies—specifically three—have directly investigated the effects of MR blockade on human MASLD and its associated hepatic fibrosis [51,52,53]. These trials offer early insights into the therapeutic potential of MR antagonists, though findings remain mixed and should be interpreted within the context of their design limitations.

A 52-week randomized controlled trial (RCT), examined the impact of low-dose spironolactone (25 mg once daily) in combination with vitamin E (600 IU/day) compared to vitamin E monotherapy in patients with biopsy-confirmed MASLD [51]. In this study, the group receiving the combination treatment demonstrated a statistically significant reduction in several noninvasive biomarkers associated with MASLD activity, as well as in HOMA-IR values, indicating an improvement in insulin sensitivity. In contrast, the monotherapy group receiving only vitamin E did not experience similar metabolic or hepatic benefits. These findings suggest that MR blockade with spironolactone may exert additive or synergistic effects when combined with antioxidant therapy in MASLD, particularly with respect to insulin resistance and noninvasive markers of hepatic inflammation and steatosis [51].

A second interventional study—a 26-week, double-blind RCT—randomized 140 patients with T2D to receive either eplerenone or placebo in order to assess potential benefits of MR antagonism on liver fat content and metabolic parameters [52]. Contrary to expectations, eplerenone did not lead to significant improvements in hepatic steatosis or metabolic outcomes in this cohort. However, it is important to note that approximately 70% of the participants were already on chronic RAAS-blocking therapy (e.g., ACE inhibitors or ARBs) at the time of randomization. This high background usage may have confounded the ability to detect any incremental benefits from MR-specific inhibition, potentially masking the therapeutic signal [52].

More recently, a third open-label RCT assessed the efficacy of two therapeutic strategies in a population of 36 adolescent girls with PCOS, a condition known to be associated with elevated RAAS activity and an increased risk of MASLD [53,54]. Participants were randomized to receive either a traditional hormonal contraceptive regimen (ethinylestradiol plus levonorgestrel) or a novel low-dose combination therapy consisting of spironolactone (50 mg), pioglitazone (7.5 mg), and metformin (850 mg)—referred to as the SPIOMET protocol. Only the SPIOMET-treated group showed normalization of visceral adiposity and fasting insulinemia, along with a significant decrease in hepatic fat content determined using MRI. In addition, treatment with SPIOMET led to measurable improvements in gut microbiota composition, suggesting possible systemic anti-inflammatory and metabolic benefits beyond hepatic endpoints [53,54].

In summary, these interventional studies exploring MR blockade in the setting of MASLD have yielded variable results. While some evidence supports a beneficial role for MR antagonists, particularly in specific subgroups or when used in combination with other agents, the overall conclusions remain inconclusive due to important methodological differences and potential biases. These include small sample sizes, heterogeneous patient populations, background RAAS inhibitor use, and differing study durations and endpoints. As such, further large-scale, well-controlled clinical trials are warranted to better elucidate the efficacy and therapeutic positioning of MR blockers in the treatment of MASLD and MASH.

## 3. The Gap of Knowledge and Evidence Regarding the Relationship Between Aldosterone and MASLD

### 3.1. Gap of Knowledge in the Mechanisms Implicating Aldosterone and MR in MASLD

The precise mechanisms and molecular pathways through which aldosterone contributes to the pathophysiology of MASLD and the development of liver fibrosis remain incompletely elucidated. While the role of aldosterone and MR activation is well established in promoting fibrosis in other organs such as the myocardium, kidneys, and vasculature, their involvement in hepatic fibrogenesis is less clearly defined. Recent studies focusing on myocardial fibrosis have shed light on the interaction between MR and aldosterone, revealing that this signaling axis leads to the downregulation of A-kinase anchoring protein 12 (AKAP12), a member of the AKAP family that is involved in coordinating various essential cellular functions, including myofibroblast differentiation, cell proliferation, and the secretion of extracellular matrix components like collagen [55]. This downregulation was associated with impaired mitochondrial biogenesis and an increase in oxidative stress within cardiac tissue [55]. Furthermore, silencing AKAP12 expression in human cardiac fibroblasts reproduced the detrimental effects observed with aldosterone exposure, while conversely, overexpression of AKAP12 was shown to counteract aldosterone-induced mitochondrial dysfunction [55]. These findings suggest that AKAP12 downregulation may represent a key molecular mechanism by which aldosterone promotes fibrotic remodeling in the heart. Intriguingly, emerging evidence also highlights a critical role for AKAP12 in the resolution of liver fibrosis, where it appears to suppress the activation of hepatic fibroblasts and potentially other non-parenchymal cells involved in fibrogenesis [56]. However, it remains unclear whether MR blockade in MASLD exerts its anti-inflammatory and antifibrotic effects through modulation of AKAP12 signaling pathways in hepatocytes or hepatic stellate cells (HSCs), as this has yet to be explored in experimental or clinical settings. In addition, no hepatic transcriptomic profiles specifically reflecting MR signaling have been described in MASLD or other liver-related disorders, whether in animal models or in human studies. This represents a significant gap in our molecular understanding. Transcriptomic analyses of aldosterone-regulated genes in other tissues—such as vascular endothelial cells, retinal tissue, and myocardial cells—have revealed divergent gene expression signatures, implying that MR signaling may exhibit distinct, organ-specific molecular patterns [57,58,59,60].

The mammalian target of rapamycin complex (mTORc), particularly mTORc2, acts as the hydrophobic motif kinase responsible for the activation of serum- and glucocorticoid-regulated kinase 1 (SGK1). This kinase is essential for the full execution of MR signaling and the expression of epithelial sodium channels (eNAC) in renal tissues [61,62]. Notably, recent studies have implicated mTORc signaling in the pathogenesis of MASLD. Activation of this pathway has been shown to drive de novo lipogenesis in the liver, in part by upregulating sterol regulatory element-binding protein 1c (SREBP-1c) transcription, a key regulator of lipid metabolism [63,64]. This process contributes not only to hepatic steatosis but also to the progression of liver fibrosis and the eventual development of hepatocellular carcinoma in MASLD. Nevertheless, it remains to be determined whether MR-mediated SGK1 activation in MASLD is dependent on mTORc signaling and whether MR directly promotes activation of the mTORc pathway in hepatic cells. These unanswered questions highlight the need for further mechanistic research to clarify the precise role of MR and its downstream pathways in MASLD-associated hepatic injury.

### 3.2. Gap of Knowledge in Human Interventional Studies

There were very few studies that specifically examined the therapeutic impact of MR blockade in the management of MASH and its associated progression to liver fibrosis. Experimental models consistently indicate that aldosterone and MR activation play a key pathogenic role in the development and advancement of steatohepatitis and fibrotic changes linked to MASLD. In these preclinical settings, MR antagonism has demonstrated a potentially beneficial effect, suggesting it could be a promising therapeutic option. However, the clinical interventional studies conducted thus far, as summarized above, were limited by their small sample sizes and the heterogeneity of the enrolled populations, which may have introduced variability in outcomes and reduced the strength of the conclusions [51,52,53]. Another important limitation is that efficacy endpoints in most of these trials were primarily based on noninvasive diagnostic markers, which are known to have relatively low sensitivity for detecting subtle changes in liver fibrosis or inflammation [52,53]. Notably, none of the studies conducted a comprehensive evaluation of the hormonal profile of the RAAS in participants undergoing MR blockade. This is a significant omission, as one could reasonably hypothesize that the effectiveness of MR antagonists may be influenced by the degree of baseline RAAS activation. Furthermore, no trials have investigated how MR blockade compares directly to lifestyle interventions, which are widely recognized as the first-line and most effective strategy for managing MASLD. This limits our understanding of the real therapeutic value of MR antagonism in this context.

## 4. Conclusions–Perspectives

Aldosterone and MR signaling appear to play critical roles in the development and progression of MASLD and MASH, extending beyond their traditional roles in fluid and electrolyte homeostasis. MR activation promotes hepatic inflammation and fibrosis via multiple molecular pathways, notably involving SGK1, TGF-β, NF-κB, and the NLRP3 inflammasome. Experimental studies strongly support the therapeutic potential of MR blockade in MASLD.

Nonetheless, substantial gaps remain in our understanding of these mechanisms, particularly in human subjects. Future research should prioritize transcriptomic and proteomic profiling of MR signaling in hepatic tissues, the elucidation of organ-specific MR signatures, and clarification of the interplay between MR and mTOR signaling. Moreover, clinical trials with well-defined patient populations, rigorous endpoint assessments, and stratification by RAAS activation status are needed to determine the true therapeutic value of MR antagonists in MASLD.

In summary, targeting aldosterone and MR signaling represents a promising but underexplored avenue in the fight against MASLD and its complications. Ongoing and future studies will be essential in validating this approach and translating preclinical insights into effective clinical interventions.

## Figures and Tables

**Figure 1 biomedicines-13-01792-f001:**
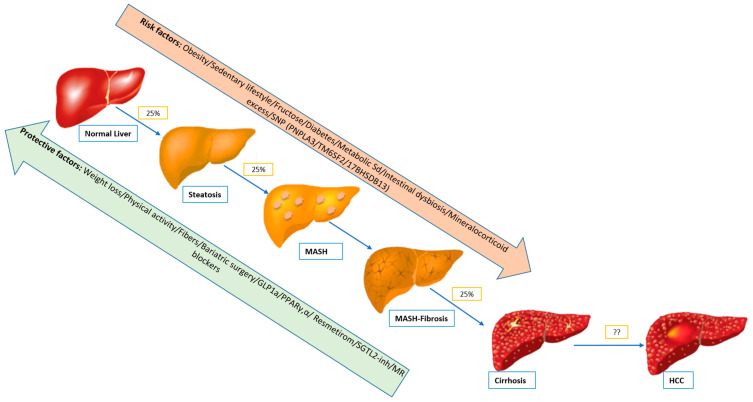
Different stages of the natural history of MASLD with protective and risk factors. Abbreviations: GLP1a: glucagon-like peptide 1 agonist, PNPLA3: patatin-like phospholipase domain containing 3, PPARϒ,α: peroxisome proliferator-activated receptor gamma and alpha, SGLT2 inh: sodium-glucose cotransporter inhibitor, STORMS: Selective Thyroid Hormone Receptor Modulators (STORMs)TM6SF2: transmembrane 6 superfamily member 2.

**Figure 2 biomedicines-13-01792-f002:**
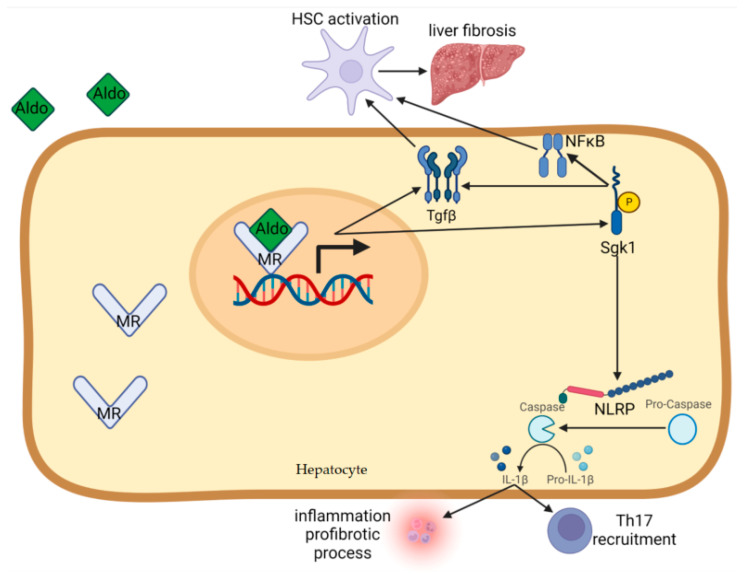
Mechanisms linking aldosterone and MR activation to MASLD: This schematic illustrates the cascade by which aldosterone, through MR binding, activates SGK1 and TGFβ signaling pathways. This leads to downstream activation of NFκB and NLRP3 inflammasome, promoting inflammation and fibrosis. The pathway also includes Th17 differentiation and IL1-β production, both contributing to hepatic fibrogenesis. Abbreviations: Aldo: aldosterone, IL1-β: interleukin 1 β, HSC: hepatic stellate cell, MR: mineralocorticoid receptor, NFκB: nuclear factor kappa B, NLRP: NOD-like receptor family pyrin domain containing 3, Sgk1: serum- and glucocorticoid-inducible kinase 1, Tgfβ: transforming growth factor β, Th17: lymphocyte Th 17.

## Data Availability

Not applicable.

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
