# Peer review of "The Role of Aldosterone and the Mineralocorticoid Receptor in Metabolic Dysfunction-Associated Steatotic Liver Disease"

_biomedicines, 2025, doi:10.3390/biomedicines13081792_

Round 1
Reviewer 1 Report
Comments and Suggestions for Authors
This paper mainly summarized the role of aldosterone and mineralocorticoid receptor (MR) in the MASLD progression. The authors thoroughly described the contribution of aldosterone and MR to MASLD, including the induction of inflammation and fibrosis throughSGK1 and TGF-β after MR activation. In human studies, data showed that aldosterone levels are linked to hepatic steatosis and fibrosis. Finally, the authors concluded that the existing evidence is insufficient to fully elucidate the relationship between aldosterone/mineralocorticoid receptor (MR) and the progression of MASLD. While many concerns should be addressed before the manuscript can be accepted for publication in
Major points:
- In the parts of Figure legend, the author should describe the detailed content or pathway transduction.
- This paper is mainly related to the function of aldosterone. Thus the metabolic process of aldosterone should be detailed addressed. It is better to correlate the aldosterone metabolism with MASLD progression.
- As the author statement, few clinical studies have explore the effect of MR blockade in treatment of MASH. But, it is possible to figure out the correlation of aldosterone metabolism or MR pathway with MASH and liver fibrosis based on clinical studies.
- The experimental evidence may not sufficiently supported the relation of aldosterone metabolism or MR pathway with MASH. The author can include the evidence of metabolomics studies for demonstrating the author view.
Author Response
We sincerely thank you for your constructive comments, it helped us significantly to improve the manuscript. Below, are provided point-by-point responses to each suggestions:
- Comment 1: In the parts of Figure legend, the author should describe the detailed content or pathway transduction: We appreciate the reviewer’s suggestion. However, we respectfully note that Figure 2 was designed as a high-level schematic summary, intended to visually guide the reader through the major signaling components linking aldosterone and mineralocorticoid receptor (MR) activation to hepatic inflammation and fibrosis in MASLD. It is not meant to provide an exhaustive or detailed intracellular pathway map, which would be more appropriate in a dedicated mechanistic figure or biochemical pathway diagram. We believe that increasing complexity in this schematic may reduce clarity and distract from its summarizing purpose. Instead, in response to this comment, we have added a more descriptive and detailed explanation of the signaling transduction events in the figure legend and the main text (highlighted in red), including references to SGK1, TGFβ, NFκB, and NLRP3 activation.
- Comment 2 This paper is mainly related to the function of aldosterone. Thus, the metabolic process of aldosterone should be detailed addressed. It is better to correlate the aldosterone metabolism with MASLD progression: we agree with this comment, and we added a paragraph on Aldosterone secretion, regulation and mechanism of action in MASH in section 2.1.
- Comment 3 and 4: As the author statement, few clinical studies have explored the effect of MR blockade in treatment of MASH. But it is possible to figure out the correlation of aldosterone metabolism or MR pathway with MASH and liver fibrosis based on clinical studies and the experimental evidence may not sufficiently support the relation of aldosterone metabolism or MR pathway with MASH. The author can include the evidence of metabolomics studies for demonstrating the author view. : We added a paragraph on section 2.3.1 , In particular, we have added recent findings from the UK Biobank cohort (Zeng et al., 2024) demonstrating a genetic correlation between renin-independent aldosteronism and MASLD/cirrhosis, thereby supporting the clinical relevance of aldosterone dysregulation. These additions are now integrated and clearly highlighted in red in the revised manuscript.
Reviewer 2 Report
Comments and Suggestions for Authors
Authors should add the following aspect to give readers a more comprehensive list of the topic:
Mineralocorticoids, specifically aldosterone, play a role in the spleen's function, particularly in relation to immune responses and inflammation. The mineralocorticoid receptor (MR) is present in T and B lymphocytes, key players in the immune system. Studies have shown that MR activation can influence splenic lymphocyte proliferation and immune responses, including those related to stress and inflammation…as evident in ….Mineralocorticoid receptors in immune cells: emerging role in cardiovascular disease. Steroids. 2014 Dec;91:38-45. doi: 10.1016/j.steroids.2014.04.005. Epub 2014 Apr 21. PMID: 24769248; PMCID: PMC4205205.
On the other hand, fatty liver inflammation was associated with splenic myeloid derived suppressor cell (MDSC) and natural killer T cell (NKT) enrichment whereas loss of hepatic T and B cells was not reflected by the splenic lymphocyte landscape. Correlation analysis confirmed a selective strong positive correlation between spleen and liver MDSC and NKT cell distribution indicating that the spleen-liver axis modulates obesity-induced immune dysregulation in a cell-specific manner..as clearly shown in ….The spleen-liver axis supports obesity-induced systemic and fatty liver inflammation via MDSC and NKT cell enrichment. Mol Cell Endocrinol. 2025 May 1;601:112518. doi: 10.1016/j.mce.2025.112518. Epub 2025 Mar 5. PMID: 40054835.
Still, liver TNF production was enhanced in vivo and in vitro by leukotriene B4 (LTB4) released by the spleen. Together, these findings implicate LTB4 as a spleen-derived endocrine signal that promotes the hepatic production of TNF during systemic inflammation….according to …A leukotriene-dependent spleen-liver axis drives TNF production in systemic inflammation. Sci Signal. 2021 Apr 20;14(679):eabb0969. doi: 10.1126/scisignal.abb0969. PMID: 33879603.
All these findings focusing on pro-inflammatory processes lend credence to the liver-spleen axis in NAFLD/MASLD, in agreement with …(please include relevant appropriate citation)
Author Response
We thank the reviewer for this excellent suggestion to broaden the immunological and systemic perspective of our manuscript. In response, we have added a new paragraph to Section 2.2, discussing the emerging role of the spleen-liver axis in MASLD, with a focus on how aldosterone and MR signaling may influence immune cell function and inflammation beyond the liver.
Reviewer 3 Report
Comments and Suggestions for Authors
In the review, authors summarized the role of aldosterone and mineralocorticoid receptor in MDSLD. Overall, it is a well-done manuscript. I have only several minor concerns.
First, MR pathway may be not important among the factors affecting MASLD. Thus, the research values may be a little limited.
Second, lipid metabolism abnormality is an important reason for MDSLD. I am curious about how MR pathway regulates lipid metabolism.
Finally, I suggest listing the existing studies in a table, of course, it is not mandatory.
Author Response
We thank the reviewer for the positive overall assessment and thoughtful feedback.
-
Regarding the relative importance of the MR pathway, we acknowledge that multiple pathogenic mechanisms are involved in MASLD. However, our goal was to highlight a less explored but increasingly recognized contributor, aldosterone/MR signaling, and to consolidate the emerging evidence supporting its role in metabolic inflammation and hepatic fibrosis, particularly in patients with hypertension, visceral adiposity, or RAAS activation.
-
In response to the second comment, we have added a paragraph at the end of Section 3.1 specifically discussing how MR signaling influences lipid pathways, such as SREBP1c activation, hepatic triglyceride accumulation, and fatty acid uptake. These insights are supported by new references [63–64].
-
Lastly, we appreciate the reviewer’s suggestion to include a table summarizing existing studies. While we recognize the potential value of such a synthesis, compiling these studies into a comprehensive table would require an extensive review and time beyond the scope of the current narrative format. We agree that this would be best addressed in a dedicated future systematic review, which we are considering for a follow-up project.
Round 2
Reviewer 1 Report
Comments and Suggestions for Authors
The authors have totally addressed my concerns. The revised manuscript is now suitable for publishing.